# An Aptamer-Based Proteomic Analysis of Plasma from Cats (*Felis catus*) with Clinical Feline Infectious Peritonitis

**DOI:** 10.3390/v16010141

**Published:** 2024-01-18

**Authors:** Benjamin E. Curtis, Zaid Abdo, Barbara Graham, Alora LaVoy, Samantha J. M. Evans, Kelly Santangelo, Gregg A. Dean

**Affiliations:** Department of Microbiology, Immunology, and Pathology, Colorado State University, Fort Collins, CO 80523, USA; bencurti@med.umich.edu (B.E.C.); alora.lavoy@colostate.edu (A.L.); samantha.evans@colostate.edu (S.J.M.E.); kelly.santangelo@colostate.edu (K.S.)

**Keywords:** feline infectious peritonitis, FIP, protein profile, proteomics, pathway analysis, enrichment

## Abstract

Feline infectious peritonitis (FIP) is a systemic disease manifestation of feline coronavirus (FCoV) and is the most important cause of infectious disease-related deaths in domestic cats. FIP has a variable clinical manifestation but is most often characterized by widespread vasculitis with visceral involvement and/or neurological disease that is typically fatal in the absence of antiviral therapy. Using an aptamer-based proteomics assay, we analyzed the plasma protein profiles of cats who were naturally infected with FIP (n = 19) in comparison to the plasma protein profiles of cats who were clinically healthy and negative for FCoV (n = 17) and cats who were positive for the enteric form of FCoV (n = 9). We identified 442 proteins that were significantly differentiable; in total, 219 increased and 223 decreased in FIP plasma versus clinically healthy cat plasma. Pathway enrichment and associated analyses showed that differentiable proteins were related to immune system processes, including the innate immune response, cytokine signaling, and antigen presentation, as well as apoptosis and vascular integrity. The relevance of these findings is discussed in the context of previous studies. While these results have the potential to inform diagnostic, therapeutic, and preventative investigations, they represent only a first step, and will require further validation.

## 1. Introduction

Type 1 feline coronavirus (FCoV-1), the most common circulating coronavirus in domestic cats worldwide, is associated with two disease biotypes [1,2,3]. The most common biotype is a localized infection of enterocytes throughout the gastrointestinal tract, which is often clinically unapparent but may cause mild enteropathy. It is hypothesized that a shift in tropism toward monocytes and macrophages allows the for systemic spread of the feline enteric coronavirus (FECV), resulting in a second biotype known as feline infectious peritonitis (FIP), a condition that is fatal in nearly 100% of untreated cases [4]. The clinical presentation of FIP can be highly variable, ranging from the presence of large volumes of pauci-cellular proteinaceous effusions within body cavities (wet form) to inflammatory nodules within various organs (dry form) [5]. As a result, the clinical signs are often not specific (e.g., weight loss, fever, malaise, and depression), and laboratory changes are variable, making the diagnosis of FIP challenging. Furthermore, diagnostic methods that are specific to FIP are hampered by low viremia and a limited ability to discriminate these two FCoV biotypes [6,7].

The complexity of FCoVs and their pathological variability has hampered the development of diagnostics, therapeutics, and vaccines. The mutation hypothesis suggests that a shift in viral tropism to macrophages/monocytes enables systemic dissemination; however, there is a poor correlation with specific viral mutations, the systemic detection of the virus, and the development of FIP [8]. Host factors likely play a significant role in allowing the systemic spread of FCoV and the variable immunopathology that is characteristic of FIP cases [9,10]. Studies exploring the immunopathogenesis of FIP at the local tissue level have reported alterations in both innate and adaptive immune responses, including increased lymphocyte apoptosis [11], suppression of regulatory T-cells [12], and alterations of cytokine levels [13]. While FIP is generally considered a systemic condition, the degree to which these specific changes occur systemically remains unclear.

Blood plasma contains thousands of proteins that are involved in various biological mechanisms, such as maintaining homeostasis, controlling inflammation, transporting molecules, and cellular communication [14]. Perturbations of metabolic, cellular, and immune pathways can be used to investigate the comparative pathology of clinically important viral infections such as FCoV. In the present study, we sought to identify proteomic similarities and differences of cats presenting with wet and dry forms of FIP compared to clinically healthy cats who were negative for FCoV and cats with active FECV infection. Protein profiles were used to query pathway databases. The results of the pathway analysis illuminate FIP’s pathogenesis and may help inform future investigations of diagnostics, therapeutics, and preventatives.

## 2. Materials and Methods

### 2.1. Patient Samples

Whole blood was collected from 52 domestic cats (*Felis catus*) in either heparin or EDTA anticoagulant tubes and centrifuged for 10 min at 2200× *g*. Separated plasma was aliquoted and frozen at −80 °C. Cats were assigned into one of four groups based on clinical presentation, FCoV screening, and necropsy findings: clinically healthy, FECV+, FIP dry form, or FIP wet form (Table 1).

### 2.2. Preinclusion Screening Methods

For cats grouped as clinically healthy and FECV+, exposure to FCoV was determined by serological antibody ELISA (FIPV-1000, IVD technologies, Inc.; Santa Ana, CA, USA). Cats grouped as clinically healthy were negative serologically and were also negative for FCoV RNA in fecal samples by real-time RT-PCR [15,16]. Cats grouped as FECV+ were defined as FCoV antibody-positive serologically and positive for FCoV RNA in fecal samples. Cats grouped into the FIP categories were FIV- and FELV-negative by SNAP test (IDEXX Laboratories, Westbrook, ME, USA), showed clinical signs that were consistent with FIP, and were shown to be positive for FCoV antigen by immunohistochemistry and/or FIPV RNA by qRT-PCR as previously described [17,18]. FIP cases were further classified as wet or dry form based on clinical or necropsy observations and history. Type 1 FCoV infection was confirmed by PCR or immunohistochemistry, as previously described [18,19]. Appendix A shows the complete patient signalment, breed, clinical history, and diagnostic tests performed. 

All FCoV cases were naturally infected; that is, no cats were experimentally infected with FECV or FIPV. Samples from FIP cases were acquired from cats presenting to the Colorado State University (CSU) Veterinary Teaching Hospital (Fort Collins, CO, USA) or the University of California Davis Veterinary Hospital (Davis, CA, USA). Samples from FCoV-negative, clinically healthy control cats and cats with active FECV infection were acquired from the CSU cat colony. FECV has historically been detected as endemic within sections of the colony. All cats with FECV were followed until they were negative for fecal FCoV RNA, and none progressed to FIP. 

### 2.3. Protein Quantification—Aptamer-Based Assay (SomaScan)

Plasma proteins were quantified using the SomaScan Assay (SomaLogic, Inc.; Boulder, CO, USA). The SomaScan assay quantifies proteins using proprietary “Slow Off-rate Modified Aptamer” (SOMAmer) reagents. SOMAmers are synthetic, single-stranded DNA sequences with protein-like modifications that tightly bind to target proteins with high specificity. The assay is validated for use with samples that present a variety of matrices, including plasma, serum, cerebrospinal fluid, and urine, among others. Following analysis, the data were normalized in three stages: Hybridization Control Normalization, followed by Median Signal Normalization, and lastly, Inter-Plate Calibration [20]. Five samples failed acceptance criteria after data normalization and were excluded from the analysis. Necropsy and immunohistochemistry results on two cats were inconclusive due to autolysis or freeze/thaw artifacts and were dropped from the analysis, leaving 45 cats in total for comparison. 

The assay utilized for this study identified 1305 unique plasma proteins. Of the unique plasma proteins measured, approximately 47% were secreted proteins, 25% were intracellular proteins, and 28% were extracellular domains. Measured protein types included, but were not limited to, cytokines, proteases, protease inhibitors, hormones, kinases, and structural proteins. Following data normalization and calibration for feline samples, one protein was dropped, leaving a total of 1304 unique proteins quantified. GraphPad Prism version 9.0 (GraphPad Software, San Diego, CA, USA) was utilized for generation of graphics for Figures 1–5.

### 2.4. Post-Assay Data Normalization, Significance Calculations, and FDR Justification

To ensure that differential changes calculated between groups would not be skewed by direction, the quantified proteins were first transformed using log base 2 (log2) calculations [21]. Significance between the various group comparisons used an unpaired Welch’s *t*-test followed by false discovery rate (FDR)-adjusted *p*-value calculation using the two-stage step-up method of Benjamini, Krieger, and Yekutieli [22]. 

With the aim of the project being to identify proteins or genes that are likely to be important in the pathology of FIP, we adjusted the FDR cutoff to 5% to reduce the risk of excluding important proteins while acknowledging the increased chance of type 1 errors. 

### 2.5. Pathway Enrichment and Analysis

Given the limited number of cats presenting with clinical FIP, as well as the variable makeup of the clinically healthy cats who were present in the CSU cat colony, age and sex matching was not possible; as such, differential analysis was also run, comparing these factors. Linear models were fit to each protein separately to evaluate differences, which flagged 8 proteins where sex was identified as a potential confounder. These 8 proteins were excluded from further analyses and are shown in Appendix A.

Names of significantly differentiable proteins identified by SomaScan were converted to UniProt, Entrez, and Ensembl gene identification (ID) numbers. In some instances, conversions resulted in non-unique IDs, such as when the plasma protein exists as a complex (e.g., HSP90α and β), fragment, or distinct activation form (e.g., C3b/iC3b). To address these instances, complexes were split into their respective components and related genes; then, the protein with the most significant adjusted *p*-value was used and the other was discarded. Splitting into individual genes resulted in 1362 IDs, of which 72 were discarded, leaving 1290 unique IDs. Appendix A show the full list of proteins and the identifications used for analyses, as well as the list of proteins that were modified or discarded.

Unique IDs, corresponding log2 fold change calculations, and 5% FDR-adjusted *p*-values were utilized in pathway analysis. The Reactome pathway browser and analysis tools (reactome.org [23]), as well as pathway enrichment software CytoScape version 3.10.1 [24], were used to query the pathway databases KEGG, Reactome, GO biological, and GO molecular to determine the most significantly enriched pathways across group comparisons (Table 3 and Appendix A). Unlike DESeq, pathway analysis has the benefit of utilizing the aggregate of proteins that have been screened, comparing pathway to pathway rather than gene/protein to gene/protein. This way, pathways which have a statistically significant number of proteins represented in them are still highlighted, even if the available data points are lower than the overall pathway size, as is typically the case with protein quantification compared to RNAseq. RStudio software 4.3.2 [25] was utilized to run the Bioconductor packages Pathview and ComplexHeatmap [26] for generation of the pathway maps (Figure 2 and Appendix A). Utilization of multiple enrichment and pathway browsing systems provided redundancy in analysis and a cross-check of enrichment results. 

## 3. Results

### 3.1. Group Comparison Overview—Differentiable Proteins

Comparisons across the four study cohorts were performed for each of the 1304 proteins that were quantified using the SomaScan assay to identify significantly different proteins between paired groups. The primary aim was to elucidate the biological responses and pathological mechanisms that were specific to cats with FIP. Comparisons between clinically healthy cats and cats with FIP revealed clear proteomic differences resulting from the FIP biotype. Additional comparisons of FIP vs. non-FIP, FIP vs. FECV, FIP wet vs. FIP dry, etc., were statistically evaluated, and the total numbers of significantly different proteins across cohort comparisons are shown in Table 2. 

The most significantly different proteins were primarily related to processes of the immune system, signal transduction, and cell survival/programmed cell death. Some of the more specific categories of proteins included cytokines and chemokines, cell signaling proteins, apoptosis-related proteins, chemotaxis and cell adhesion molecules, and growth factors. Differentiable proteins with a log2 fold change greater than 2.0 or less than −2.0 are highlighted in Figure 1. Figure 2 highlights proteins which effectively cluster cats with FIP from clinically healthy cats. Interestingly, five cats in the FIP group did not cluster with the majority of cats with FIP or completely clustered with the clinically healthy cats. The difference in these five cats could not be accounted for by patient signalment (age, breed, or sex and reproductive status) or by FIP manifestation (wet form versus dry form). The complete list of proteins is available in the Appendix A. 

### 3.2. Feline Enteric Coronavirus vs. Clinically Healthy Cats

Of the 1304 proteins quantified, none were significantly different between FECV and clinically healthy groups. Forty-six proteins had a *p*-value < 0.05 but failed to pass the 5% FDR cutoff (Figure 3C). FIP vs. FECV had 445 proteins with an FDR < 0.05, 16 with a log2 FC < −2, and 19 with a log2 FC > 2. FIP vs. clinically healthy cats had 442 proteins with an FDR < 0.05, 14 with a log2 FC < −2, and 18 with a log2 FC > 2. Comparing significantly different proteins between FIP vs. FECV and FIP vs. clinically healthy revealed a large degree of overlap. Thirteen proteins with a log2 FC < −2 were present in both comparisons, and 16 proteins with a log2 FC > 2 were present in both comparisons. In total, 353 proteins (~80%) of significant proteins were present in both comparisons (Figure 3A–D, Appendix A).

### 3.3. Significantly Different Interleukins: FIP vs. Clinically Healthy

Given the characteristic inflammatory lesions that are associated with FIP, we sought to determine the systemic levels of cytokines that were involved in the incitement and control of inflammation. The SomaScan assay quantified 138 proteins that were categorized as cytokines [27]. Interestingly, there was a significant decrease in pro-inflammatory mediators including IL-1b, IL-5, IL-6 sRa, IL-8 (CXCL8), IL-17RA, RANTES (CCL5), CSF-1 (M-CSF), prostaglandin COX-2, and CD23 and increases in anti-inflammatory mediators TNF sR-II, IL-10Ra, IL-2 sRg, STAT3, heat-shock proteins (HSP70 and 90), and CD36. Figure 4 highlights the interleukins which were significantly different between cats with FIP and clinically healthy cats. 

### 3.4. Key Biological Systems Affected by FIP: Pathway Enrichment and Analysis

Significantly different proteins across cohort comparisons were interrogated against multiple pathway databases and analysis tools. Consistent with the predominant protein classes that were identified among the most differentiable proteins, the enrichment analyses identified pathways related to the immune system, signal transduction, cell growth and activation, and programmed cell death as those that were most over-represented by the enrichment analysis. The enrichment of a pathway does not necessarily signify functional or directional action by that pathway, i.e., many pathways contain both the activating as well as the inhibitory factors that mitigate the signaling and effector functions within a given pathway. Directional differences between cohorts must be considered to determine the biological outcomes. With the aim to extract more relevant information, differential expression (log2 fold change) as well as the significance in difference (FDR < 0.05) were used to rank the most significant proteins and thereby curate the pathways that were most affected by the clinical condition, FIP. A selection of the pathways explored here is shown in Table 3. The full list of enriched pathways is available in the Appendix A.

#### 3.4.1. General Immune System Pathways

Of the 442 differentiable proteins, nearly half (202 proteins) were related to the immune system, with 115 being significantly higher and 87 significantly lower. Broadly, the immune system-related pathways can be divided into three main branches, the innate immune system, the adaptive immune system, and cytokine signaling within the immune system. In total, there are 113 proteins mapped to the innate immune system (70 higher and 43 lower) and 62 proteins mapped to the adaptive immune system (45 higher and 17 lower). Some proteins are represented in both innate and adaptive pathways. Cytokine signaling within the immune system was highly represented, with 108 significant proteins (55 higher and 53 lower). To better elucidate the effects of FIP infection compared to clinically healthy cats, the more targeted significantly enriched pathways downstream from these three main branches were explored.

#### 3.4.2. Pattern Recognition Receptor (PRR) Pathways

Pathways related to two classes of PRRs, toll-like receptors (TLRs) and C-type lectin receptors (CLRs), were identified by enrichment analyses in cats with FIP. Specifically, DC-SIGN, ICAMs, and DC-SIGNR were involved in enriching CLR receptor-related pathways, while TLR4, TAK1-TAB1, and TLR4:MD-2 complex were among the proteins that were involved in enriching TLR-related pathways.

Many viruses have evolved to subvert the effects of PRR signaling and/or utilize PRR as receptors/coreceptors for infection [28,29]. Similarly, the family of cathepsin proteins have also been utilized by several viruses for host cell entry [28]. Here, we quantified several cathepsins with significantly increased levels of expression in FIP cats compared to clinically healthy cats. Specifically, cathepsin B and D were significantly increased, while cathepsin G and S were increased but failed to pass the 0.05 FDR cutoff for significance.

#### 3.4.3. Signaling Pathways—Mitogen-Activated Protein Kinase (MAPK), Janus-Kinase/Signal Transducers and Activators of Transcription (JAK/STAT), and Natural Killer (NK) Cell Activation/Inhibition

Two protein tyrosine phosphatases (PTP6/SHP-1 and PTP11/SHP-2) that negatively regulate JAK-STAT signaling were significantly increased in cats with FIP, while JAK2 (JAK 1 and 3 were not measured) was significantly lower. No difference between clinically healthy cats and cats with FIP could be detected for STAT1 and STAT6. Interestingly, there were markedly increased levels of STAT3 in cats with FIP. STAT3 can be phosphorylated independently of JAKs by receptor-associated kinases like Tyk2 (increased) and MAPK kinases like p38 MAPK (significantly increased). Elevated STAT3 activity can lead to the upregulation of inhibitory receptors, such as killer cell immunoglobulin-like receptors (KIRs), which dampen NK cell activation and effector functions. Additional evidence that supports NK cell inhibition in cats with FIP included the increased levels of the NK inhibitory receptors KIR2DL4 (significantly increased) and KIR3DL2 (Table 4). The KEGG pathway analyses for MAPK, JAK-STAT, and NK cell cytotoxicity are shown in Appendix A [25,26]. 

#### 3.4.4. Apoptosis

The differentiable proteins between cats with FIP and clinically healthy cats identified multiple apoptosis-related pathways. Specifically, there was enrichment in programmed cell death, apoptosis, the intrinsic pathway for apoptosis, and the activation of the BCL2-associated agonist of cell death (BAD) and translocation to mitochondria pathways. Table 5 highlights the differentiable proteins and their roles as either pro- or antiapoptotic.

#### 3.4.5. Vascular Integrity

For cats with wet form FIP, there were several significantly different proteins related to controlling vascular permeability. VEGF and several of its types and isoforms were significantly different in wet form FIP cats compared to clinically healthy cats (Figure 5).

## 4. Discussion

Unravelling the pathogenesis of FIP has historically relied on gross and microscopic characterization, cellular analysis, or targeted molecular investigation in tissues or in vitro systems. Experimental infections and in vitro studies are often limited to the much less prevalent Type 2 FCoV. We postulated that the power of omics approaches would provide information that might accelerate our ability to diagnose, treat, and prevent fatal FIP resulting from Type 1 FCoV. We employed a novel plasma proteomics approach to gain insight at a systemic level. We identified differentiable proteins, their genes, and associated biological systems and pathways to illuminate the pathogenesis of FIP and identify key questions for future investigations. 

### 4.1. Application of SomaLogic Aptamer-Based Proteomics in Cats

There were several advantages to using the SomaScan assay for proteomic analysis of feline plasma. The assay is highly multiplexed, sensitive, and quantitative and requires only 150 µL of sample to measure over 1300 proteins. The modified DNA aptamers (SOMAmers) were selected by the manufacturer to bind native proteins with slow dissociation off-rate kinetics, thus providing greater sensitivity for the detection of low-abundance proteins [30]. In the SomaScan assay, SOMAmer–protein complexes are captured on beads, and unbound proteins are removed by washing. The complexes are then released and subjected to a polyanionic competitor that displaces non-cognate proteins by binding the SOMAmer. The complexes are then recaptured on beads, and the unique SOMAmers are quantified to provide an average median dynamic range of 4.2 logs. This approach greatly reduces the likelihood of non-specific interactions; however, given that the assay has been developed for human samples, the interpretation of results from non-human species must be carefully considered. Proof-of-principle for application of the assay to non-human primates, rat, mouse, dog, and cat has been shown by the manufacturer [20]. The present study is the first experimental application in cats, with much more work performed with murine samples [20,31]. 

An interpretation of results from cats using the SomaScan assay must acknowledge that some proteins may not be sufficiently conserved to achieve binding of the human optimized SOMAmer. Further, the sensitivity of the assay may be impacted by interactions that are weaker but not abrogated by structural differences between species. Comparing cats across treatment groups mitigates some of these issues, since differences in the relative amount of a particular protein may still be meaningful, even if the affinity of the SOMAmer is lower for the cat protein. The interpretation of an individual protein must be approached with caution, particularly when seeking to identify biomarkers for a particular condition. In that case, other assays (e.g., ELISA) should be employed for confirmation. An analysis of biological pathways considers multiple proteins in aggregate to determine whether a pathway might be affected by a treatment or disease condition, and therefore, it reduces the potential for a single protein to influence the interpretation. Pathway analysis also provides the advantage of revealing the biological impact of increased or decreased levels of individual proteins of interest. Ultimately, using a proteomics platform that has not been fully validated for cats provides only a first step, and results must be further validated. 

To the authors’ knowledge, this is the first application of protein analysis performed at the systemic level via plasma for cats presenting with clinical FIP. Previous publications regarding pathway analysis or differential expression profiles in cats with FIP have primarily focused on the gene expression within cell culture or tissue samples [19,32,33,34,35,36]. Furthermore, many of these studies used Type 2 [19,32,33,34,35,36] or lab-adapted strains of Type 1 viruses [19,34]. 

### 4.2. Feline Coronavirus—FIPV vs. FECV

Several mutations have been associated with the transition of Type 1 FCoV from FECV to FIPV [4,18,37,38,39,40,41,42]. FIPV is considered the etiology of both wet and dry form feline infectious peritonitis, while FECV is a minimally pathogenic pathotype [43,44,45,46]. Our results here highlight this pathological dichotomy, with 442 significantly different proteins (FDR < 0.05) between cats with FIP and clinically healthy cats and 0 significant proteins (FDR < 0.05) between cats with FECV and clinically healthy cats (Figure 3). This finding is consistent with the clinically silent presentation that typifies FECV infection and demonstrates the capacity of the enteric coronavirus to replicate without stimulating a systemic host response.

### 4.3. FIP Systemic Cytokine Response 

Cytokines are integral to immune cell activation, recruitment, and the initiation of effector functions [47]. Some of the predominant cytokine profiles previously described for FIP include increases in pro-inflammatory cytokines such as IL-1β and IL-6, as well as decreases in IL-4 and IL-2. There has been mixed reporting for changes in TNF, IL-10, and IL-12. Most of these studies evaluated changes at the tissue level, such as within lymph nodes or other organs with clinical lesions, or utilized cell cultures with lab-adapted strains of virus and measured mRNA levels rather than protein levels [13,48,49,50,51,52]. The findings of the proteomic and pathway analyses identified some similar patterns, but as is highlighted in Figure 4, there were also differences at the systemic level. 

Expected results included increased inflammatory mediators and signaling proteins such as leukotrienes and cell adhesion molecules that direct leukocyte chemotaxis and neutrophil degranulation. Other cytokines, including IFNγ and TNF, that might have been expected to be increased based on previous tissue-level studies, were not significantly different compared to clinically healthy cats, and the potent pro-inflammatory cytokine, IL-1β, was significantly lower. This suggest that while FIP is remarkably inflammatory at the site of infection within tissues, cats with FIP are able to regulate pro-inflammatory signals systemically and avert a cytokine storm as has been described for some coronaviruses [53] and hypothesized as a function of FIP [54]. A 2006 study evaluating cytokine mRNA levels within whole blood samples of cats with FIP (n = 3) also found insignificant differences in IL-1β, IL-6, and TNF. The authors concluded that larger studies would be needed to verify those results [29]. With modestly larger sample sizes, our results support those of Gelain et al. [29]. 

### 4.4. Pattern Recognition Receptors

The pathway enrichment analysis highlighted systemic effects of FIP on the innate immune system and cytokine/interleukin signaling. Central to the initiation of the innate immune response is the activation of PRRs by pathogen-associated molecular patterns (PAMPs) or damage-associated molecular patterns (DAMPs), with subsequent triggering of signaling pathways such as the MAPK pathway and activation of transcription factors like NF-κB [55]. Pathways related to two classes of PRRs, TLRs and CLRs, were identified by enrichment analyses in cats with FIP. 

DC-SIGN is an important CLR that is present primarily on dendritic cells and, when activated, can stimulate signaling of the MAPK pathways, NF-κB pathway, and type I interferon regulation. DC-SIGN has been identified as a potential coreceptor for Type 1 FCoV infection [19,56]. In humans, the DC-SIGN-related protein (DC-SIGNR) is a homolog of DC-SIGN and recognizes many of the same ligands as DC-SIGN but has a broader range of cell expression that includes endothelial cells [57,58]. Consistent with previous findings, we identified higher levels of DC-SIGN and, for the first time, significantly increased DC-SIGNR in cats with FIP. This observation brings together the common immunohistochemical association of FCoV antigen within macrophages as well as endothelial cells.

RNA viruses are typically recognized by endosomal TLR3, 7, and 8. TLR2 and 4 reside on cell surface plasma membranes but may also play a role in viral infections through the recognition of viral proteins [59]. Studies related to FIP have described a mixed expression picture for TLR3, 7, and 9. Using in vitro infection, monocytes from seropositive cats had decreased levels of TLR7 expression, and uninfected monocytes had increased levels of TLR7, leading the authors to hypothesize that TLR7 might be a key factor in the viral evasion of the innate immune response [48]. Tissue-level studies using clinical FIP cases identified significant increases in the expression of TLR2, 4, and 8 in the mesenteric lymph nodes [60]. These results were similar to the plasma level measurements that we noted here, with significantly increased TLR2 and slight increases in TLR4 for cats with FIP compared to clinically healthy cats. The TLRs quantified by the SomaScan were limited to TLR2, 4, and the TLR4:MD-2 complex, which limits overall conclusions. However, it is interesting to note that use of DC-SIGN as a coreceptor by SARS-CoV-2 down-modulates the function of TLR4 in human dendritic cells [61]. Similar relationships remain to be investigated in FIP.

### 4.5. Signaling Pathways

The biological goal following the recognition of PAMPs by PRRs is the activation of the immune system to control the source of the PAMPs, generally via the activation of signaling pathways and the production of proteins such as pro-inflammatory cytokines that can further aid in the control of infection. Two of the most significant signaling pathways that were flagged by enrichment analysis were the MAPK and the JAK-STAT signaling pathways. 

In a simplified scheme, the MAPK proteins signal through three primary pathways: the ERK1/2, p38, and JNK pathways [62]. The extracellular-signal-regulated kinase (ERK) pathways are primarily activated by growth hormones, while the c-Jun N-terminal kinases (JNK) and p38 MAPK pathways are primarily activated by stress or inflammatory signals, such as cytokines [62]. The p38 arm of the MAPK pathway has been postulated to be the mechanism by which FIPV induces the strong inflammatory response which characterizes the typical FIP lesions [63]. Our findings here seem to support p38 as the primary MAPK pathway that is initiated in cats with FIP. Proteins related to the ERK and JNK pathways were not significantly differentiable (Appendix A), and these pathways were not highlighted by pathway enrichment.

The enrichment analysis identified the JAK-STAT signaling pathway as significant. The JAK-STAT pathway is utilized by many cells to convert the cytokine binding to cell surface receptors into transcription signals within the nucleus that upregulate inflammatory and immune mediators [64]. There are currently four known JAKs and seven known STATs [65]. The SomaLogic 1.3 panel utilized here measures JAK2 and Tyk2, as well as STAT 1, 3, and 6. STAT1 plays significant roles in both innate and adaptive immune responses [64,66,67] and can act as a signal transducer to all major interferon types [67,68,69]. STAT6 is primarily involved in Th2 polarization [64], and STAT3 has been shown to have dual roles, regulating both pro- and anti-inflammatory states as well as Th2 and Th17 polarization [64]. STAT1 and -6 were not differentiable within the plasma of cats with FIP compared to clinically healthy cats, but STAT3 was significantly increased. JAK2 was significantly decreased, but STAT3 can be phosphorylated independently of JAKs by receptor-associated kinases like Tyk2 (increased) and MAPK kinases like p38 (significantly increased). Recent studies in human patients with clinical COVID-19 have suggested the dysregulation of JAK-STAT signaling as a contributor to elevated pro-inflammatory cytokines in circulation [70]. This suggests that further exploration of JAK-STAT signaling in cats with FIP might inform therapeutic interventions. 

### 4.6. Apoptosis Pathways

Lymphopenia is a common clinical finding in cats with FIP. This has been observed both in whole blood samples via the complete blood count and locally within affected lymph nodes histologically. Studies investigating this phenomenon have suggested lymphocyte apoptosis as the cause [71,72]. At the systemic level, we found that multiple pathways that are related to apoptosis were significantly upregulated in cats with FIP. The pathway enrichment of BAD’s activation and translocation to the mitochondria suggests that signaling via the intrinsic pathway may be the mechanism by which clinical FIP causes lymphopenia. 

### 4.7. Vascular Integrity—Wet FIP 

VEGF was originally designated vascular permeability factor [73], highlighting its role in vascular integrity. The development of effusion in wet form FIP has been hypothesized to be a result of antibody–antigen–complement complexes that affix to vessel walls leading to vasculitis, decreased vascular integrity, and leakage [74]. However, both the wet and dry forms of FIP are histologically characterized by marked vasculitis [45], and only the former produces effusions. Later work attributed the vasculitis to activated, pro-inflammatory macrophages [75] and the production of VEGF [76]. There are currently five known types of VEGF (A, B, C, D, and placental growth factor). VEGF-A additionally has multiple isoforms, such as VEGF121 [77]. Previous reports have described higher plasma levels of VEGF in experimentally infected cats that developed wet form FIP [76]. We similarly measured increased plasma levels of VEGF (VEGF-A) in cats with FIP. Notably, only cats with the wet form of FIP had significantly elevated VEGF levels, suggesting that this protein may be key in the progression from vasculitis without effusion to the development of effusion.

### 4.8. The Viral Evasion of Immune Response

Major histocompatibility complex class-I (MHC-I) plays a critical role in presenting foreign intracellular peptides to cytotoxic T-cells. As such, it is not surprising that several viruses have evolved mechanisms to undermine this process. Human papillomavirus has been shown to downregulate components of MHC-I antigen presentation such as TAP-1 [78], consequently reducing the detection of the virus by host immune cells [79]. Another key protein in this system, calreticulin, plays a role in facilitating the loading of peptides and stabilizing MHC-I [80]. In cats with FIP, we detected lower levels of calreticulin, suggesting a possible role in undermining MHC-I antigen presentation. Additionally, while ubiquitination is often involved in processing antigens for presentation, polyubiquitin protein K63 is involved in the endocytosis of the MHC-I complex and the facilitation of its degradation in endosomes [81]. Viral proteins of the Kaposi’s sarcoma herpes virus have been shown to induce this ubiquitin [82]. Cats with FIP showed significantly higher levels of K63 polyUb, suggesting a possible role by which the virus may manipulate MHC-I to evade immune detection. 

Canonically, cells with reduced expression of MHC-I should be targeted for destruction by NK cells. It has previously been reported that cats with FIP have reduced NK cell function [12]. Some viruses, such as human papillomavirus, are able to evade NK cells despite suppressing MHC-I [79,83]. The pathway analysis suggests that cats with FIP might have reduced NK cell-killing function (Table 4, Appendix A). Elevated STAT3 activity can lead to the upregulation of inhibitory receptors, such as KIRs that dampen NK cell activation and effector functions [84]. There are increased levels of the NK inhibitory receptors KIR2DL4 (significantly increased) and KIR3DL2 (increased), along with the previously mentioned proteins SHP-1 and SHP-2 (PTPN6 and 11, both significantly increased), whose downstream effector activities include the inhibition of NK cell function [7,79,85,86]. MICA, a stimulatory receptor for NK cell activation, was also significantly decreased in cats with FIP. This combination of proteins which can reduce MHC-I expression, along with a protein profile indicating suppressed NK cell activity, may explain how FIP avoids NK-cell antiviral responses.

### 4.9. Concluding Thoughts

This cross-sectional study evaluated plasma protein levels from a single time point for each cat enrolled. It is likely that the protein quantities and/or gene expression may differ at the tissue level compared to the systemic level reflected in the blood, as well as at different time points throughout the course of the disease. However, an advantage of this approach is that the physiological levels of proteins could be measured, avoiding the uncertainty of extrapolating from mRNA levels, which may not correlate temporally to protein levels [87,88]. Proteomics and pathway analysis are tools that can be used for hypothesis generation, hypothesis testing, and more targeted exploration to facilitate the development of novel diagnostics, preventatives, and therapeutic modalities. Direct protein measurements performed in this study allow for the discovery of candidate biomarkers of FIP. Current diagnostics for FIP are often invasive (histopathology/immunohistochemistry) or struggle to differentiate the two biotypes of feline coronavirus (FIPV vs. FECV) [6,7]. The lack of clustering of all FIP cases observed in the heatmap (Figure 2) using the significant proteins with the greatest log fold change suggests that biomarker discovery may need to assess multiple biomarker candidates longitudinally during disease progression. Investigations using orthogonal assays are ongoing to determine the clinical value of proteins that consistently differentiated FIP cats from non-FIP cats. 

## Figures and Tables

**Figure 1 viruses-16-00141-f001:**
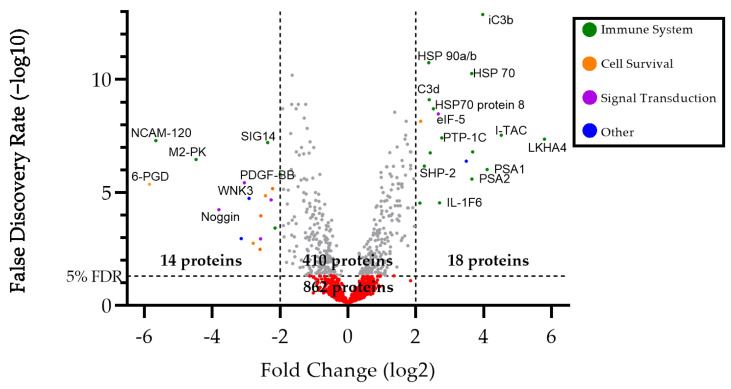
Volcano plot comparing FIP vs. clinically healthy cat groups. In total, 18 proteins had a log2 fold change (FC) > 2.0, and 14 proteins had FC < −2.0. Proteins above the 5% FDR line were considered statistically differentiable. Only proteins with a FC > |2.0| are highlighted. The most differentiable proteins are primarily grouped into functional categories related to the immune system, cell survival, and signal transduction.

**Figure 2 viruses-16-00141-f002:**
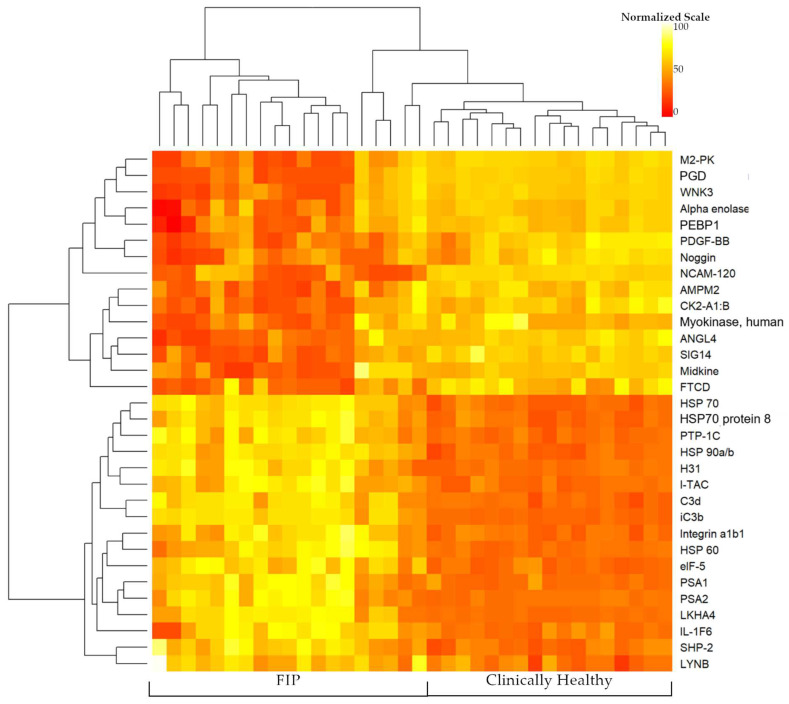
Heatmap comparing FIP vs. clinically healthy cats. Quantified proteins are log2-adjusted and scaled from 0 to 100 to allow for comparing relative changes. Hierarchical clustering was applied to both cats and the top 30 proteins with the highest fold change (the full protein list can be seen in Appendix A). (Rstudio, Bioconductor, ComplexHeatmap).

**Figure 3 viruses-16-00141-f003:**
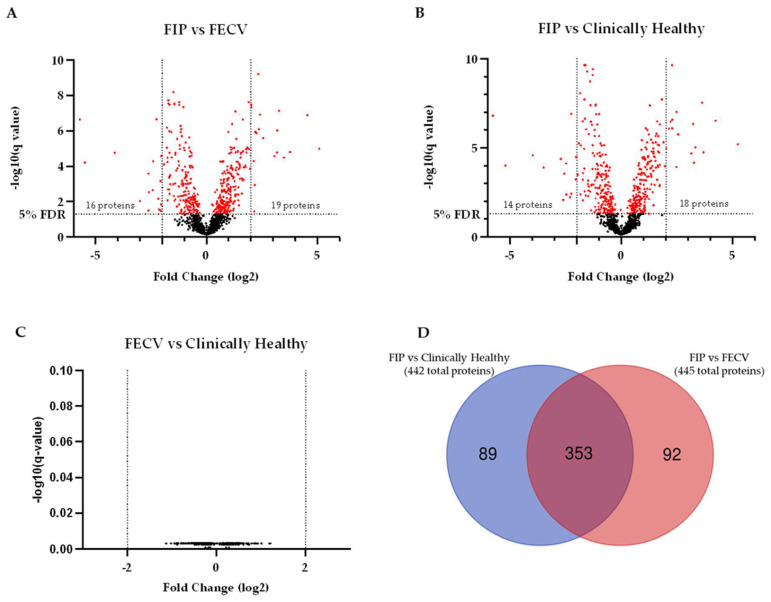
Volcano plots and Venn diagram comparing clinically healthy, FIP, and FECV groups. (**A**) Cats with FIP compared to cats with FECV; (**B**) cats with FIP compared to clinically healthy cats; (**C**) cats with FECV compared to clinically healthy cats; (**D**) overlap of proteins between the FIP comparisons and clinically healthy and FECV cats.

**Figure 4 viruses-16-00141-f004:**
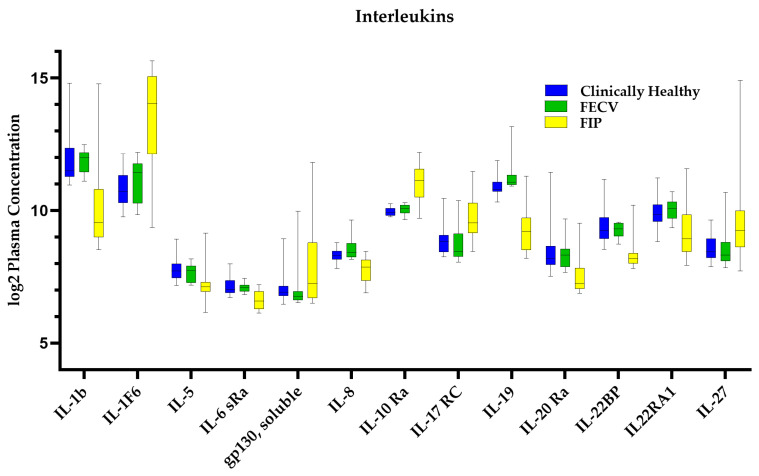
Significantly different interleukins. Boxplots are shown for clinically healthy, FIP, and FECV groups. Due to the large number of interleukins quantified, only those which were significant are shown. All other interleukins and cytokines that were quantified can be seen in Appendix A.

**Figure 5 viruses-16-00141-f005:**
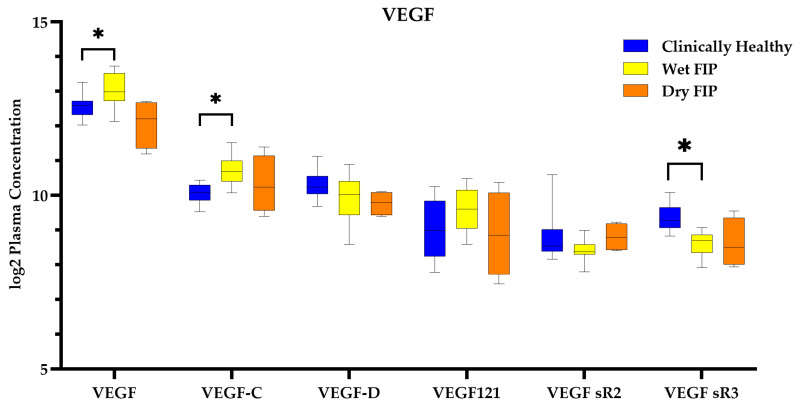
Boxplots of all VEGF-related proteins quantified by the SomaScan. The asterisk (*) denotes proteins that are significantly different at FDR < 0.05. Clinically healthy cats compared to cats with wet form FIP and cats with dry form FIP.

**Table 1 viruses-16-00141-t001:** Distribution of cats tested: sex and reproductive status, FIV/FELV and FCoV status.

Clinical Group (n)	Sex (n)	FIV/FELV	FCoV
Clinically Healthy (17)	Female (12) Neutered Male (5)	negative	negative
FECV+ (9)	Female (6) Neutered Male (3)	negative	positive
FIP dry (4)	Spayed Female (1) Neutered Male (3)	negative	positive
FIP wet (15)	Female (1) Spayed Female (3) Male (1) Neutered Male (10)	negative	positive

**Table 2 viruses-16-00141-t002:** Number of significant proteins between group comparisons.

Comparison	Significant Proteins at 5% FDR
FIP (n = 19) vs. Clinically Healthy (n = 17)	442
FIP (n = 19) vs. FECV (n = 9)	445
FIP wet form (n = 15) vs. FIP dry form (n = 4)	2
FECV (n = 9) vs. Clinically Healthy (n = 17)	0

**Table 3 viruses-16-00141-t003:** Selected significant pathways—cats with clinical FIP compared to clinically healthy cats. (Reactome and KEGG databases).

Category	Pathway ID	Pathway Description	FDR
Pattern Recognition Receptor (PRR)-Related Pathways	R-HSA-5621481	C-type lectin receptors (CLRs)	0.00497
	R-HSA-937072	TRAF6-mediated induction of TAK1 complex within TLR4 complex	0.00411
	R-HSA-166016	Toll-Like Receptor 4 (TLR4) Cascade	1.95 × 10^−4^
Signaling Pathways	R-HSA-9006934	Signaling by Receptor Tyrosine Kinases	3.63 × 10^−14^
	hsa04010	MAPK signaling pathway	4.70 × 10^−4^
	hsa04630	JAK-STAT signaling pathway	2.14 × 10^−8^
	R-HSA-5673001	RAF/MAP kinase cascade	1.13 × 10^−6^
	R-HSA-199418	Negative regulation of the PI3K/AKT network	3.45 × 10^−6^
Immune Pathways	R-HSA-6798695	Neutrophil degranulation	2.44 × 10^−7^
	R-HSA-5668541	TNFR2 non-canonical NF-kB pathway	3.23 × 10^−6^
	hsa04650	Natural killer cell-mediated cytotoxicity	3.54 × 10^−5^
	R-HSA-1236975	Antigen processing—cross presentation	7.26 × 10^−5^
	R-HSA-9018676	Biosynthesis of D-series resolvins	0.0346
	R-HSA-9018677	Biosynthesis of DHA-derived SPMs	0.0208
Apoptosis	R-HSA-5357801	Programmed cell ceath	5.69 × 10^−8^
	R-HSA-109606	Intrinsic pathway for apoptosis	3.95 × 10^−6^
	R-HSA-111447	Activation of BAD and translocation to mitochondria	1.06 × 10^−5^
Vascular Integrity-Related Pathways	R-HSA-194138	Signaling by VEGF	5.59 × 10^−7^
	R-HSA-5218920	VEGFR2-mediated vascular permeability	0.00751

**Table 4 viruses-16-00141-t004:** Natural killer cell-related proteins, their functional direction, and their quantified direction. Quantified proteins with FDR > 5% are shown below the grey bar. See Appendix A for full log2 fold change and significance measurements.

Protein	Stimulatory/Inhibitory	Increased/Decreased
SHP-1 (PTPN-6)	Inhibitory	Increased
SHP-2 (PTPN-11)	Inhibitory	Increased
KIR2DL4	Inhibitory	Increased
STAT3	Inhibitory	Increased
MICA	Stimulatory	Decreased
	FDR > 0.05	
KIR3DL2	Inhibitory	Increased
Tyk2	Inhibitory	Increased

**Table 5 viruses-16-00141-t005:** Apoptosis-related proteins, their functional direction, and their quantified direction. Quantified proteins with FDR > 5% are shown below the grey bar. See Appendix A for full fold change and significance measurements.

Protein	Pro-/Antiapoptotic	Increased/Decreased
BAD	Pro-apoptotic	Increased
BCL2-L1	Antiapoptotic	Decreased
Cytochrome C	Pro-apoptotic	Increased
Caspase 10	Pro-apoptotic	Increased
AKT 1, 2, & 3	Antiapoptotic	Decreased
Calcineurin	Pro-apoptotic	Increased
	FDR > 0.05	
iAP proteins (BIRC-3, 5, & 7)	Antiapoptotic	Decreased
Caspase 2 & 3	Pro-apoptotic	Increased
BID	Pro-apoptotic	Increased
Granzyme B	Pro-apoptotic	Increased

## Data Availability

The complete repository of data presented in this study can be found in the Appendix A.

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
