# Peer review of "An Aptamer-Based Proteomic Analysis of Plasma from Cats (Felis catus) with Clinical Feline Infectious Peritonitis"

_viruses, 2024, doi:10.3390/v16010141_

Round 1
Reviewer 1 Report
Comments and Suggestions for Authors
The authors present results from a proteomics investigation of cats with FIP, cats with the enteric form of FCoV, and FCoV-uninfected/clinically healthy cats. The manuscript is well written and I think the results are important?
I put a question mark because I honestly don’t know. I am a clinical veterinarian with a PhD in virology and serology, and I found the manuscript challenging to read and hard to draw meaningful conclusions from. Make no mistake, this may be due to my own inadequacies rather than a reflection of the quality of the study or of the manuscript. But if the authors could spell out the importance a little more of their findings for someone like me, I think it will improve the engagement and readability of the study.
For example – they state their findings have ‘the potential to inform diagnostic, therapeutic, and preventative investigations.’
Did they look at this in their work? And if not, perhaps they could look at this? For example, could a FIP diagnostic algorithm be constructed from the ‘219 increased and 223 decreased [proteins] in FIP cases’?
Or could they better argue how their findings might specifically help therapeutic or preventative measures? Are there examples from other diseases they could refer to?
Perhaps they will argue that this is for future studies to determine. But if they can help address my ineptitude, I think it would dramatically strengthen the value of their manuscript.
The headings in the Discussion suggest the authors were trying to argue the significance of their findings, but it is so long and difficult to read that any meaning was lost on me. Perhaps the authors can rewrite/shorten/simplify slightly?
Additional points below. Most importantly I would like much more information on test results from each cat to give me confidence that infection categories have been assigned correctly!
Also 11 figures and 5 tables is far too much to take in – can the authors please choose the most important and halve this amount?
Can the journal please assist the authors with referencing so that multiple references are contained within the one set of brackets, not multiple separate sets of brackets.
L15 (and L62) – I think you should make it clear that the ‘clinically healthy’ cats were FCoV-negative
L38 – ‘symptoms’ is a human term. Please replace with ‘clinical signs’
L42 – change referencing to [6,7] (no gap between)
L43 – no apostrophe in FCoV’s
L47, 49, 295, 415, 430, 476, 481, 482 - I don’t like and think stating ‘reviewed in’ is unnecessary/is clunky – consider deleting
L77 – Is ‘Pearson 2019’ meant to be a reference? This looks like an error
L81 – change to [16, 17]
L83 – change to [17, 18]
L75-90 – The findings from this study hinge on the case definition. I know the paper is already overly long, but given case definitions is absolutely fundamental I think this section demands more information. Perhaps a supplementary table with each animal and precisely the diagnostic tests used to assign each cat to a category? Importantly I would like to know how the authors were confident that none of the enteric FCoV-infected cats were in the early stages of FIP?
Table 1 – I found confusing since the 45 study animals are duplicated – are the last 2 rows really necessary? Can they be deleted?
Figures 7, 8 , 9C, 11 – The differences between groups don’t look that impressive to me? Seems to be a lot of overlap?
Figures 2 and 10 – I don’t know how to read a heat map. Can you summarise any pertinent findings in the Figure legends?
Reviewer 2 Report
Comments and Suggestions for Authors
In the presented study, Curtis and co-authors used a commercial aptamer based approach to quantify changes in cat serum. The authors are interested in detecting any changes between healthy cats and cats with FIP or the enteric form of FCoV. The authors identified a large number of changes at the protein level for cats with FIP compared to healthy or FECV. Additionally, they even identified 2 differentially affected proteins with the wet vs dry form of FIP.
I congratulate the authors on identifying these changes which may provide useful to develop a clinical tool for quick diagnosis.
However, I believe the manuscript could be substantially improved.
1) Experimental design: The authors included details of the study groups of cats used in the study, including sex, indicating that they are aware that such characteristics can make a significant difference to serum composition.
Therefore I would like to suggest that further characteristics e.g. breed of cats should be included, in particular if there is a difference between healthy individuals and infected cats.
In addition, there is a mayor shortcoming in the study as the cats in the comparison are not gender matched. I do appreciate that clinical symptomatic cats e.g. dry vs wet FIP are hard to match, however this should not be relevant for the healthy individuals. Non-FIP has 8/26 male and FIP contains 14/19 males. Therefore it seems entirely possible that any changes detected could be mainly based on differences in gender. I would strongly encourage the authors to include more males in their clinically healthy group and reanalyze the data with these additional datapoints.
2) According to the text there are 265 significantly different proteins across all comparisons. However, this does not seem to fit with table 2 which lists over 400 significant proteins. Can the authors please clarify which numbers are correct. It seems different stages of the analysis might have been used.
3) The heatmap in figure 2 shows a very nice pattern differentiating between FIP and control animals. In my mind this may be the main finding of the paper. However, the authors hardly discuss this in the text. For example, even though there is a strong differentiation between FIB and control, there are 5 FIP samples which does not quite fit the trend and lie in between the two groups. Do the authors have any potential explanations for these outliers?
Additionally, the graph would benefit from labelling which part contains FIP and control samples rather than relying on the reader to identify this by the names on the bottom.
4) The conclusions of this manuscript are solely based on the initial aptamer assay. However, the proteins outlined in figure 2 might reveal clinical relevant markers. The manuscript would be significantly strengthened if the authors could confirm these biomarkers by an orthogonal approach e.g. ELISA or western blot analysis of serum samples for at least a selection of proteins.
5) In Figure 3, the authors display the volcano plot for the individual comparisons. It would be useful if they could include a Venn diagram showing the overlap between the more than 400 proteins identified in the comparisons in panel A vs B. In particular, are the higher confidence hits overlapping.
6) Figure 3 panel C does only show that there are no significant hits and is probably a distraction. They authors may consider removing this panel.
7) In table 3 and 4 the authors try to identify particularly important pathways. However, from the text, I am not 100% sure that the correct comparisons are drawn. For example, the aptamer approach used can identify up to 1300 proteins. However for enrichment analysis the pathway size for immune system is 1956 with 115 significant genes.
It stands to reason that only a fraction of the 1300 proteins tested are within this superset. So 115 proteins out of that much smaller number should be the more relevant comparison e.g. how many proteins with the classification immune system are within the test set. If all are, 115 would not be of any interest, whereas if only 200 are, this would be very important. Can the authors clarify how the analysis was performed and correct for the limited test set used in the aptamer approach.
I expect the FDR-corrected p values will consequently change significantly which may affect downstream discussion and subsequent figures.
A clear example of this problem is given by the authors in 3.4.3. 400 proteins were quantified by the SomaScan assay as related to cytokine signalling pathways. 100 of these are significantly changing, so ¼. However overall >400 proteins change in the whole testset of ~1300, so roughly 1/3. Therefore cytokine signalling has fewer changes than the rest of the dataset, suggesting that it might not be as important as many of the other pathways.
8) I do appreciate that the authors are trying to make sense of this large scale dataset, which is often one of the problems of omics approaches. However, even though selected pathway analysis and clustering of changing proteins can be a useful tool, I believe the authors have gone to far here and the interpretation of the data appears more of a literature review. Unfortunately, I believe this is distracting from the key message of the manuscript which is in my mind the identification of key markers in the serum.
Whether this is due to lysis of specific cells and therefore release of marker within these cells or via any active participation in signaling pathways seems unclear to me at this point. Therefore, the extensive description of finding from the literature seems not helpful. I would suggest a dramatic shortening of this part, maybe highlighting a few key findings.
I believe that this could really facilitate the readability and usability of the manuscript for the anticipated readership.
Round 2
Reviewer 1 Report
Comments and Suggestions for Authors
Well done to the authors on the revised version of their manuscript. Greatly improved. No issues detected.
Good luck with their future studies! I hope this research translates to some improvements in the feline diagnostic and therapeutic space for FIP.
Reviewer 2 Report
Comments and Suggestions for Authors
After this round of revision, I do believe that the authors have improved the manuscript considerably, both regarding in clarifications as well as readability.
Personally, I believe it is a shame that the authors decided to remove the original figure 2 rather than improving it by adequate labelling, as to me this was the most compelling figure, clearly showing variability in the different animals while highlighting a clearly distinct pattern. I would encourage the authors to include this figure in the final manuscript to ensure their work has the most impact.
More problematic is that the authors decided not to perform any validation by orthogonal approaches. I do appreciate that independent validation in a non model organism is always difficult, but believe that confirming even a few of the targets by any of the initially suggested methods would benefit the scientific soundness of the work. The current work is relying entirely on one assay without any further validation of the results.
Finally, I would ask the authors to remove the "Here we provide evidence that" [...] have reduced killing NK cell killing function" in line 738 as this is not correct and can mislead the field. The authors provide evidence that there are some markers in the blood relating to this function, but as far as I can see have not performed any assay to asses this directly in this work.
